# COFCA: A STEP-WISE COUNTERFACTUAL MULTI-HOP QA BENCHMARK

**Jian Wu**[1*] **Linyi Yang**[2*] **Zhen Wang**[1] **Manabu Okumura**[1] **Yue Zhang**[3†]

[1]Institute of Science Tokyo [2]University College London
[3]School of Engineering, Westlake Univeristy

## ABSTRACT

While Large Language Models (LLMs) excel in question-answering (QA) tasks, their real reasoning abilities on multiple evidence retrieval and integration on Multi-hop QA tasks remain less explored. Firstly, LLMs sometimes generate answers that rely on internal memory rather than retrieving evidence and reasoning in the given context, which brings concerns about the evaluation quality of real reasoning abilities. Although previous counterfactual QA benchmarks can separate the internal memory of LLMs, they focus solely on final QA performance, which is insufficient for reporting LLMs' real reasoning abilities. Because LLMs are expected to engage in intricate reasoning processes that involve evidence retrieval and answering a series of sub-questions from given passages. Moreover, current factual Multi-hop QA (MHQA) benchmarks are annotated on open-source corpora such as Wikipedia, although useful for multi-step reasoning evaluation, they show limitations due to the potential data contamination in LLMs' pre-training stage. To address these issues, we introduce a Step-wise **Co**unter**fa**ctual benchmark (CofCA), a novel evaluation benchmark consisting of factual data and counterfactual data that reveals LLMs' real reasoning abilities on multi-step reasoning and reasoning chain evaluation. Our experimental results reveal a significant performance gap of several LLMs between Wikipedia-based factual data and counterfactual data, deeming data contamination issues in existing benchmarks. Moreover, we observe that LLMs usually bypass the correct reasoning chain, showing an inflated multi-step reasoning performance. We believe that our CofCA benchmark will enhance and facilitate the evaluations of trustworthy LLMs.

## 1 INTRODUCTION

Retrieval-augmented generation (RAG) (Chen et al., 2023; Ma et al., 2023; Gao et al., 2023b), has garnered increasing research attention (Tang & Yang, 2024; Poliakov & Shvai, 2024). As shown in Figure 1a, given a query, a RAG system first retrieves pieces of evidence from a set of relevant passages, and then generates a response by taking the passages as additional context to LLMs. While most existing work concentrates on the setting where the answer can be derived from a single retrieved passage, there are situations where LLMs are required to integrate information from multiple resources as evidence to infer the answer. As shown in Figure 1b, given the query, the single retrieved passages cannot offer enough information to generate correct answers. Instead, the evidence *"Earth is the third planet"* and *"Mars is after Earth"* are required to be integrated for deciding *"The fourth planet of the sun"*.

Recently, a line of work (Wang et al., 2023b; Staniszewski et al., 2023; Trivedi et al., 2022) has investigated the evidence integration capabilities of LLMs combined with RAG, using existing Multi-hop QA (MHQA) datasets such as HotpotQA (Yang et al., 2018) and 2WikiMultihopQA (Ho et al., 2020). However, such work focuses only on the correctness of the final answer, without evaluating the reasoning process itself. It is thus difficult to understand whether shortcut reasoning exists (Tang et al., 2021; Ho et al., 2022; Yang et al., 2022) and whether such reasoning is robust

---

[*]Equal contribution. Jian Wu did this work during his internship at Westlake University
[†]Corresponding author.

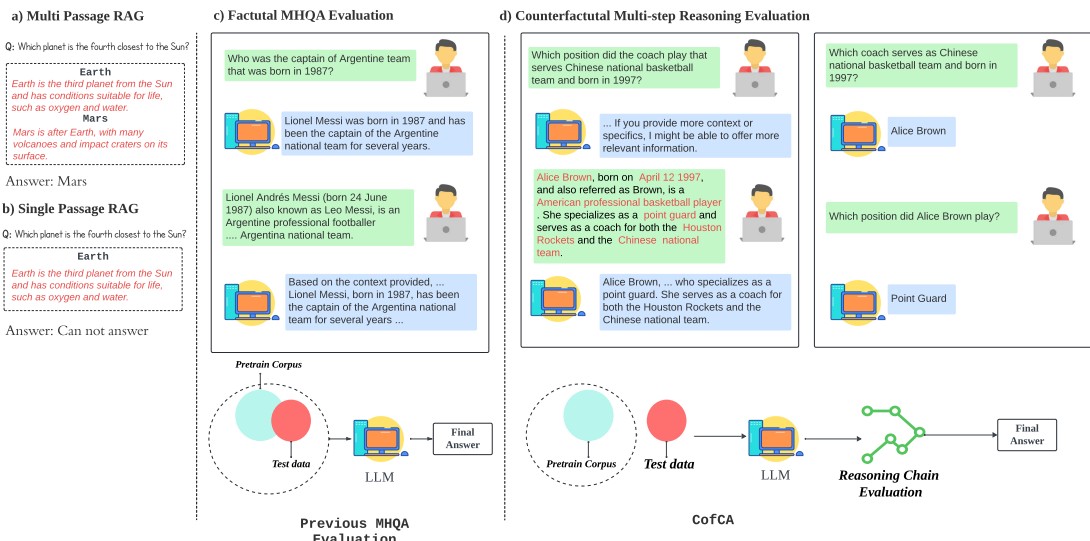

Figure 1: Differences between (a) and(b). The words in red are the pieces of evidence for the given questions. The differences between factual MHQA (c) and counterfactual MHQA (d). c: we input a factual MHQA with and without context into ChatGPT. ChatGPT could output the correct answer based on its internal memory regardless of the context. d: When inputting a counterfactual QA, where the passage is rewritten from the existing factual passage in c (words in the red), ChatGPT cannot rely on its memory and must reason on the given context, deeming that counterfactual QA can decouple LLMs' internal memory and reasoning abilities.

across out-of-distribution (OOD) tasks. Existing research (Tang et al., 2021) shows that strong QA models such as DFGN (Xiao et al., 2019), DecompRC (Min et al., 2019), and CogQA (Ding et al., 2019) can correctly answer multi-hop questions but bypass the correct reasoning chain. However, no existing research has evaluated the performance of LLMs to the same problem.

We aim to address the above issues by making a systematic evaluation of the LLMs' reasoning process. To this end, an existing sub-question style benchmark (Tang et al., 2021) derived from HotpotQA Yang et al. (2018) can be useful, in which LLMs are required to answer a series of sub-questions to reach the target rather than giving a final answer solely. However, existing datasets cannot be directly used for reporting LLM real performance, due to the risk of data leakage in the pretraining stage (L'opez et al., 2024; Zhou et al., 2023a; Fu et al., 2023). As shown in Figure 1c, for instance, given the question "Who was the captain of Argentine team that was born in 1987?", ChatGPT might give the correct answer even without a context, since relevant knowledge is learned by the model during the pre-training stage. Therefore, a sub-question benchmark that avoids the negative influence of data leakage is needed. To this purpose, we consider the counterfactual evaluation method (Li et al., 2023; Neeman et al., 2023; Zhou et al., 2023b; Wu et al., 2024; Yu et al., 2023). We make modifications to existing knowledge in Wikipedia, deriving new passages and ensuring that knowledge in these passages is non-existent in the real world, thus those answers have little chance of being inherent in LLMs. For example, in Figure 1d, given the question "Which position did the coach play that serves Chinese national basketball team and born in 1997?", LLMs cannot rely on their memories and must answer the first sub-question "Which coach serves as Chinese national basketball team and born in 1997?", before finding the final answer by asking a second sub-question "Which position did Alice Brown play?". The passage in Figure 1d, is modified from the passage in Figure 1c. Specifically, we generate counterfactual data to Wikipedia knowledge by replacing key information (named entities, noun phrases, and synonyms) and then paraphrasing (back translation).

We call our dataset CofCA, a step-wise and **Co**unter**fa**c**tu**al MHQA benchmark, which consists of counterfactual passages with corresponding multi-hop QA pairs and sub-QA pairs, and an equal amount of factual MHQA data from HotpotQA, 2WikiMultihopQA and MuSiQue as the control group. Using CofCA, we evaluate and report LLMs' real reasoning ability. Experimental results show two types of inflated performance in LLMs: 1) an obvious performance gap between Wikipedia-based

factual MHQA benchmarks and counterfactual MHQA data; 2) inflated performance due to a low proportion of correct reasoning chains as well as a high proportion of incorrect reasoning chains, with GPT-4 achieving only 36.3% correct reasoning chains across the entire dataset. Additionally, we observe that incorporating sub-questions into the prompt as part of the reasoning chain is a more efficient approach for improving LLMs' performance.

To the best of our knowledge, we are the first to introduce counterfactual data into the evaluation of the multi-step reasoning ability of LLMs, finding that there is a significant performance gap between LLMs' performance on factual data and counterfactual data. Furthermore, the reasoning abilities of LLMs are exaggerated because of the poor reasoning chain performance. The Whole CofCA data are available at `https://anonymous.4open.science/r/LLM_inherent_multi_step_eval-3818/`.

## 2    RELATED WORK

**Retrieval Augmented Generation** (RAG) improves LLM's response (Borgeaud et al., 2021) and also mitigates the occurrence of hallucinations, thereby enhancing the models' credibility (Gao et al., 2023a). As demonstrated by Khattab et al. (2021), designs a RAG system for MHQA and claim verification tasks. These tasks require the extraction of evidence from two or more documents to produce a correct answer. Tang & Yang (2024) propose a Multi-hop RAG benchmark, which consists of a large collection of multi-hop queries, ground-truth answers, and the corresponding supporting evidence. Multi-hop RAG requires LLM to reason and answer multi-hop queries given the evidence. However, LLMs' memorized knowledge sometimes conflicts with the given context, emphasizing the importance of correcting LLMs' generations with new facts. Li et al. (2023); Neeman et al. (2023); Zhou et al. (2023b); Wu et al. (2024); Yu et al. (2023) propose counterfactual QA benchmarks to separate LLMs' parametrical knowledge (internal) and contextual knowledge (outer) that fix LLMs to reasoning on the given context strictly by editing the contextual information or prompts. Previous work motivates us to explore LLMs' real reasoning ability by reasoning in counterfactual contexts. However, counterfactual QA datasets still only assess final QA performance and lack reasoning process evaluation.

**Multi-hop QA Datasets** Multi-hop QA requires more than one reasoning step in multiple paragraphs to answer a question (Dua et al., 2019; Yang et al., 2018; Ho et al., 2020; Trivedi et al., 2021). Notably, Tang et al. (2021) introduce a human-validated sub-question dataset derived from the HotpotQA dataset (Yang et al., 2018), undertaking a detailed investigation of the models' capabilities to reason through sub-questions. Their findings revealed that notable models like DFGN (Xiao et al., 2019), DecompRC (Min et al., 2019), and CogQA (Ding et al., 2019) exhibit deficiencies in resolving sub-questions, even though they may successfully address the overarching multi-hop question. Moreover, Wikipedia-based MHQA datasets face the challenge of data contamination that hard to objectively and truthfully evaluate the reasoning ability of LLMs. Data contamination, i.e., the presence of test data from downstream tasks in the training data of large language models (LLMs), is a major issue in measuring LLMs' real performance on other tasks. For example, HotpotQA (Yang et al., 2018), 2WikiMultihopQA (Ho et al., 2020), and MuSiQue (Trivedi et al., 2021) can be applied to evaluate the multi-step reasoning performance of LLMs. Typically, evaluating LLMs in MHQA datasets involves using RAG to retrieve and reason over context with a single step of retrieval. However, single-step retrieval can result in insufficient context retrieval for complex questions, as it provides a limited scope of information (Gao et al., 2023b). Prasad et al. (2023) proposes a framework that defines good reasoning chains in Correctness and Informativeness to illustrate whether the previous reasoning step could help the current reasoning step and the final answer.

**Benchmarking Data Leakage** A handful of recent studies have provided several strategies, methods, and benchmarks for detecting contamination without the need to access pre-training data (Shi et al., 2023; Roberts et al., 2023; Golchin & Surdeanu, 2023; Zhu et al., 2023a). Ravaut et al. (2024) surveys recent work on detecting data contamination and releases a python library named *LLMSanitize* that implements major contamination detection methods. However, these data contamination detection benchmarks are required to dynamically update because of the development of LLMs and the expansion of pertaining data. Dynamical maintenance is time-consuming and effortless, while our proposed benchmark CofCA, based on the knowledge edition, is fixed and maintains the cleanness of the test data. To statistically and quantitatively detect the LLMs' data contamination extent, Xu et al. (2024b) proposes a detection pipeline by computing perplexity and N-gram accuracy to evaluate

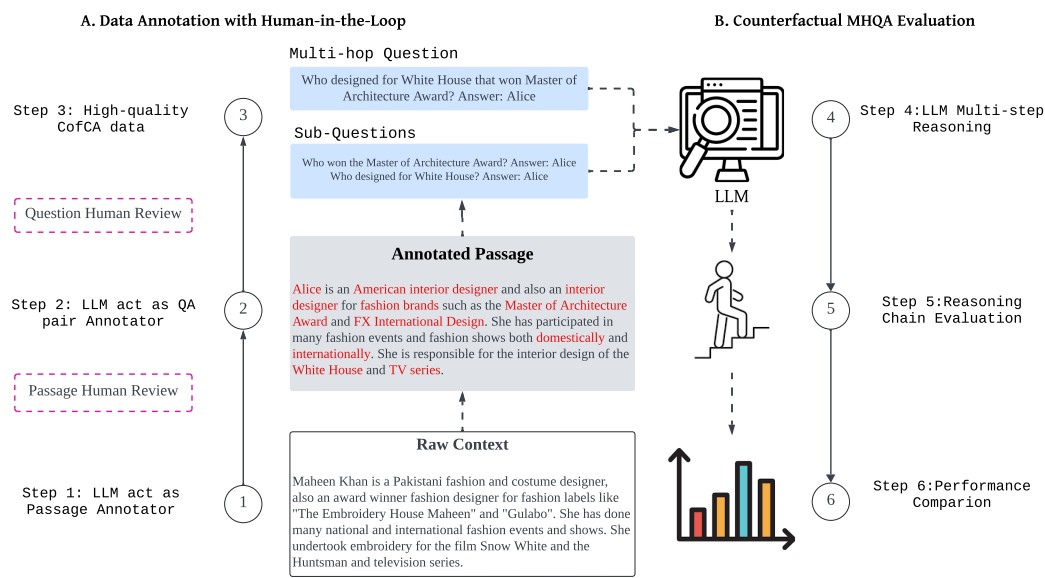

Figure 2: The framework of our LLM automatic data annotation pipeline. From left to right, **A**: we first ask LLM to act as a passage annotator to replace the keywords and paraphrasing. Then we manually ensure that the correctness of grammar and the key information have been changed. We send the reviewed high-quality data to GPT-4 to generate QA pairs and manually check the quality. **B**: After receiving the reviewed high-quality counterfactual QA data, we evaluate LLMs on generated data to test their inherent ability on MHQA.

potential data leakages. Zhang et al. (2024) designs a new Grade School Math 1000 (GSM1k) to mirror the GSM8k benchmark (Cobbe et al., 2021) and evaluate LLMs' mathematical reasoning ability. Clean-Eval (Zhu et al., 2023b), a benchmark to assess the inflated performance of LLMs. The experimental results show a drop in performance for GPT-3.5 and Llama-2 in the Clean-Eval data, deeming the data contamination problem. Xu et al. (2024a) proposes a detection pipeline with the help of perplexity and N-gram accuracy gap between the previous MathQA dataset and synthesized data, pinpointing the potential data leakage problem. The measurement illustrates that underscores the urgency for an evaluation paradigm shift in how we approach the development and evaluation of LLMs. To properly and objectively investigate multi-step reasoning abilities, it is crucial to disentangle LLMs' inherent memory from their reasoning capabilities. Hua et al. (2024) introduces a novel benchmark for counterfactual QA tasks by editing six common reasoning schemes in the real world, while still lacking muti-step reasoning evaluation for LLMs. Our work is different from such works in two main aspects: 1) a comprehensive reasoning chain evaluation; and 2) Counterfactual Question Answering that reduces the risk of data contamination.

# 3 MULTI-HOP REASONING EVALUATION

In this section, we design a human-in-the-loop annotation framework and our novel evaluation method. As shown in Figure 2, the framework consists of three components: 1) we first ask LLM such as GPT-4 to act as a passage annotator that rewrites passages from Wikipedia with human evaluation and feedback; 2) LLM's automatic counterfactual QA pairs annotation with human evaluation and feedback; 3) After obtaining the synthesized counterfactual MHQA data, we use counterfactual multi-hop reasoning to evaluate several strong LLMs to report the real multi-step reasoning performance.

## 3.1 DATA CONSTRUCTION

**Data Collection and Passage Rewriting**  We randomly select 300 Wikipedia passages from Huggingface[1] as the raw context. Inspired by recent studies on LLM's ability to aid human annotation (Bartolo et al., 2021; Törnberg, 2023), we design a pipeline for automatically annotating Wikipedia passages into counterfactual passages. Given a raw Wikipedia passage, LLMs are required to act as a passage annotator to do the named entity, noun phrase, and synonym replacement. Since LLMs are pre-trained on the open-source corpus collected from the Internet, we manually search the key information of the annotated passage on the Internet, e.g. the characters, times, events, causes, processes, and results.

After the replacement stage, we translate the replaced text into Chinese and finally back translation into English, e.g., the words in red of the annotated passage in figure 2 are the replaced named entities, noun phrases, and synonyms, and the new counterfactual passage is rewritten from the original Wikipedia passage. Upon acquiring the new and counterfactual passages, human experts conduct a manual evaluation of data quality by focusing on two key aspects: 1) assessing the grammatical integrity of the annotated passages to ensure there are no grammatical errors; and 2) verifying that the annotated passages are new and counterfactual to Large Language Models (LLMs).

**QA-pair Annotation and Checking**  We use LLMs to generate new multi-hop questions to fit the rewritten passages. To make sure the generated QA pairs are correct and related to the given passages, we check the QA pairs from two perspectives: 1) Grammar issue; 2) Answerabilities of the generated question, whether the question is related to the passage and make sure the answer can be reasoned from the passages based in the generated questions. To evaluate LLMs' performance on different complexity of multi-hop questions, for each passage, we annotate 3 complex questions: one for 2-hop, one for 3-hop, and one for 4-hop questions. LLMs are required to generate multi-hop questions along with the corresponding sub-questions for reasoning chain evaluation. For example, at the middle part of Figure 2 illustrates the re-annotated context, newly generated multi-hop questions, sub-questions, and intermediate answers. The annotated answers are all with a short answer span that follows the settings of HotpotQA, 2WkimultihopQA, and MuSiQue (Yang et al., 2018; Ho et al., 2020; Trivedi et al., 2021). To ensure the evidence of answers are integrated from multiple paragraphs, we follow the settings of HotpotQA, 2WkimultihopQA, and MuSiQue, the supporting facts are from at least two paragraphs. Here we use EM and F1 scores to measure the LLMs' output. After obtaining the rewritten passages with corresponding QA pairs and sub-qa pairs, we also check the logic of the whole passage and QA pairs. We expect that the passage is coherent and the answers can be reasoned from the passage based on the given questions. The prompts of the data annotation and more annotated examples are shown in Appendix B.

**Dataset Analysis and Statistics**  Table 1 shows the statistics of our dataset. We re-annotated 300 unique Wikipedia passages, with three multi-hop questions (one 2-hop, one 3-hop, and one 4-hop for each passage), a total of 3600 unique QA pairs including 900 multi-hop questions, and 2700 sub-questions with corresponding answers. Besides, we also randomly selected 900 factual MHQA data as the control group (300 from HotpotQA, 300 from 2WikiMultihopQA, and 300 from MuSiQue) and 1,800 data of CofCA in total. Following the settings of previous LLM evaluation benchmarks (Wang et al., 2023a; 2024), we treat the total of 1800 data as the test set. Following benchmarks such as HotpotQA (Yang et al., 2018), 2WikiMulthopQA (Ho et al., 2020), we propose a taxonomy on fine-grained question types and examples commonly used in multi-hop QA illustrated in Table 11. In HotpotQA, 2WikimultihopQA, and MuSiQue datasets, the multi-hop questions consist of two types of questions, or the combination of the two types of questions, *Bridge* and *Comparison*: *Bridge* question is required to find the bridge entity that connects the sub-questions, while *Comparison* question is a type of question that compares two or more entities for the parallel sub-questions. We focus on these two types of questions for annotating multi-hop questions.

**Inter Human Agreement**  Following a thorough manual inspection of the annotated data, we randomly select 300 samples, dividing them equally among 2-hop, 3-hop, and 4-hop instances, with each sample being evaluated by two expert reviewers. These reviewers are tasked with identifying and eliminating any data that significantly deviates from the established guidelines. Initially, the

---

[1]https://huggingface.co/datasets/lucadiliello/english_wikipedia

Table 1: Statistics of our CofCA dataset.

| 2 hop dataset | | 3 hop dataset | | 4 hop dataset | | Whole data property | Value |
|---|---|---|---|---|---|---|---|
| 2 hop QA pairs | 300 | 3 hop QA pairs | 300 | 4 hop QA pairs | 300 | Unique Passages | 1800 |
| Sub-QA pairs | 600 | Sub-QA pairs | 900 | Sub-QA pairs | 1200 | Total QA pairs | 4500 |
| Correctness | 94 | Correctness | 93 | Correctness | 92 | Sentences per data (Avg) | 38.42 |
| Informativeness | 85 | Informativeness | 86 | Informativeness | 82 | Inter-annotator Agreement | 94% |

Table 2: Differences between our CofCA with previous factual and counterfactual QA benchmarks.

| Benchmarks | Data Type | Data Source | Task | Reasoning Chain |
|---|---|---|---|---|
| HotpotQA | Factual | Wikipedia | Multi-hop/final-QA | ✗ |
| 2WikiMultihopQA | Factual | Wikipedia | Multi-hop/final-QA | ✗ |
| MusiQue | Factual | Wikipedia | Multi-hop/final-QA | ✗ |
| DisentQA | Counterfactual & Factual | Natural Questions | Single-hop/final-QA | ✗ |
| IfQA | Counterfactual & Factual | Crowdsourcing | Open Domain/final-QA | ✗ |
| **CofCA(Ours)** | Counterfactual & Factual | Rewriting Wikipedia | Multi-hop/final-QA/sub-qa | ✓ |

experts focus on assessing the grammatical correctness of the questions, sub-questions, sub-answers, and answers within the annotations. Subsequently, they verify the originality of the content in the annotated passages by checking for their presence on the Internet.

According to our statistics, the agreement rate between the annotators in the randomly selected CofCA samples is 94% and the human agreement rates are 96%, 92%, and 95% in the 2-hop, 3-hop, and 4-hop datasets, respectively. This suggests that our synthesized instances reflect good data quality on annotation guideline following and achieve high human agreement among expert annotators. To quantitatively illustrate data quality, we also utilize GPT-4 to assign scores to each selected data from two perspectives: correctness and informativeness. Each data is assigned with 1 or 0, which means correct or incorrect, informative or not informative. Correctness indicates whether the answer can be reasoned from the given question and context, while informativeness means whether the QA pairs and context are related or not.

## 3.2 EVALUATION METRICS

Given a set of input documents, we employ three representative QA evaluation methods to assess the correctness of LLM-generated MHQA responses: sub-question answering evaluation, reasoning chain evaluation, and the joint performance of sub-qa and MHQA.

**Sub-QA Evaluation** This part is the basis of all evaluation results. Following reading comprehension (Rajpurkar et al., 2016), evaluation is conducted through lexical matching using two widely used metrics to assess the performance of models. We employ $F_1$ and EM scores to evaluate the answers to sub-questions, similar to the single-hop QA task.

**Reasoning Chain Evaluation of Multi-hop QA** Table 2 illustrates the differences between our evaluation method and previous evaluation methods of counterfactual QA and factual MHQA datasets. To interpret the behavior of existing LLMs on each hop of the reasoning process required for multi-hop questions, and to determine their reasoning ability to answer simple questions, we followed the experiment setting proposed by (Tang et al., 2021). For example, in the 2-hop dataset, each data contains a 2-hop question, 2 sub-questions, 2 intermediate answers, and a final answer. In order to understand whether LLMs can correct answers by following the right reasoning chain, we calculate the proportion of right and incorrect reasoning chains to compare LLMs' reasoning performance. Each question or sub-question has two results, correct or incorrect, thus an N-hop question with its N sub-questions has $2^{(N+1)}$ different reasoning chains. Due to the space limitation, we measure and collect correctness statistics for the 2-hop question dataset, $q_{sub1}$, $q_{sub2}$, and $q$, and show the percentage of 8 reasoning chains given by LLMs.

**Joint Performance** Previous MHQA benchmarks are traditionally evaluated on the EM or $F_1$ score on the final answer (Rajpurkar et al., 2016; Yang et al., 2018; Ho et al., 2020), which is partially correct. The previous MHQA systems and LLMs are treated as a black box and we can not figure out

how they find the final answer. To understand the impact of sub-qa on MHQA, we introduce a joint performance that combines the evaluation of Sub-QA performance and MHQA performance. The details of computing the joint scores are shown in Appendix C.

Table 3: Performance of Proprietary LLMs and Open Source LLMs on Wikipedia-based factual multi-hop QA datasets. The performance is measured by EM and F1 scores with a zero-shot setting. PM† indicates the partial match of LLMs' outputs evaluated with GPT-4-turbo with the same prompt.

| Datasets | Wikipedia | | | | | | | | |
|---|---|---|---|---|---|---|---|---|---|
| | HotpotQA | | | 2Wiki | | | MuSiQue | | |
| Metrics | EM | F1 | PM † | EM | F1 | PM † | EM | F1 | PM † |
| *Proprietary LLMs* | | | | | | | | | |
| GPT-4 | 69.9±1.5 | 82.3±1.3 | 74.8 ±1.2 | 59.7±1.4 | 67.4±2.7 | 64.8±0.9 | 57.3±1.9 | 65.4±2.9 | 63.9±1.4 |
| GPT-3.5 | 58.6±0.9 | 69.1±1.1 | 62.8±0.7 | 56.3±0.9 | 67.6±0.8 | 59.4±0.9 | 49.3±0.8 | 63.2±1.5 | 53.1±0.6 |
| GEMINI-pro | 58.2±1.3 | 68.4±1.3 | 63.5±0.9 | 48.5±1.6 | 58.5±0.9 | 54.7±1.2 | 41.3±1.5 | 54.5±0.7 | 46.9±1.3 |
| text | 50.3±0.9 | 61.4±0.8 | 54.9±0.8 | 42.3±1.4 | 53.9±1.4 | 46.7±0.7 | 40.2±0.9 | 51.0±1.5 | 44.6±0.4 |
| Bing Chat | 68.1±0.6 | 78.3±1.2 | 72.1±1.2 | 58.9±0.5 | 69.9±0.5 | 63.4±0.8 | 49.6±1.1 | 64.1±0.8 | 52.3±0.7 |
| O1-preview | 72.2 ±0.6 | 82.7 ±0.7 | 76.9 ±1.1 | 68.7 ±0.9 | 79.8 ±0.8 | 72.3 ±1.2 | 63.9 ±0.9 | 72.4 ±0.5 | 67.9±0.8 |
| *Open Source LLMs* | | | | | | | | | |
| Llama 2-7b | 34.5±1.2 | 41.3±1.1 | 38.5±0.3 | 30.6±1.1 | 34.7±1.1 | 33.8±0.9 | 31.7±0.8 | 35.6±1.2 | 34.2±0.9 |
| Mistral-7b | 30.6±1.5 | 37.2±1.4 | 34.9±0.5 | 27.4±0.6 | 29.8±0.9 | 31.4±0.8 | 25.2±0.7 | 28.9±0.8 | 29.2±0.7 |
| Qwen 2-7b | 36.2±1.5 | 43.5±1.3 | 39.3±0.4 | 31.7±1.0 | 35.8±0.8 | 36.8±0.5 | 28.2±1.1 | 31.2±1.2 | 33.5±0.4 |

Table 4: Results on CofCA, the results reveal that LLMs show an obvious performance gap between previous Wikipedia-based factual data and our CofCA.

| Datasets | CofCA | | | | | | | | |
|---|---|---|---|---|---|---|---|---|---|
| | 2-hop | | | 3-hop | | | 4-hop | | |
| Metrics | EM | F1 | PM † | EM | F1 | PM † | EM | F1 | PM † |
| *Proprietary LLMs* | | | | | | | | | |
| GPT-4 | 53.1±1.5 | 62.8±1.3 | 57.6±1.1 | 44.5±1.3 | 56.4±1.7 | 49.5±1.2 | 42.3±0.6 | 53.5±0.9 | 48.8±1.1 |
| GPT-3.5 | 40.6±0.7 | 56.7±0.5 | 43.7±0.9 | 37.7±0.5 | 50.9±1.1 | 42.1±1.3 | 32.5±1.2 | 44.6±0.8 | 36.2±1.1 |
| GEMINI-pro | 35.0±0.7 | 45.3±1.6 | 38.2±0.8 | 29.6±0.5 | 42.7±0.9 | 31.9±0.6 | 26.1±1.1 | 35.3±1.2 | 29.8±1.1 |
| text | 32.6±0.9 | 48.5±0.8 | 37.4±0.9 | 27.8±0.9 | 46.3±0.8 | 33.5±1.2 | 24.8±0.8 | 44.1±0.9 | 27.6±0.7 |
| Bing Chat | 41.9±0.8 | 53.4±0.9 | 45.4±0.9 | 39.6±1.1 | 49.4±1.2 | 43.7±0.7 | 30.7±0.9 | 42.2±0.7 | 35.6±0.7 |
| O1-preview | 59.4 ±0.4 | 68.5±0.3 | 63.9±0.8 | 52.3 ±0.5 | 66.1±0.4 | 58.4±0.7 | 50.7 ±0.4 | 62.3±0.6 | 53.2±0.7 |
| *Open Source LLMs* | | | | | | | | | |
| Llama 2-7b | 26.1±0.9 | 34.3±1.2 | 30.5±0.4 | 22.6±1.1 | 26.7±1.3 | 25.8±0.8 | 24.9±1.2 | 28.7±1.1 | 30.5±0.9 |
| Mistral-7b | 24.7±0.9 | 29.5±0.7 | 28.5±0.3 | 20.8±1.1 | 25.3±1.1 | 24.8±0.7 | 18.6±1.1 | 22.2±1.3 | 23.5±0.5 |
| Qwen 2-7b | 30.8±1.1 | 38.2±1.4 | 34.1±0.4 | 27.2±1.1 | 31.7±1.3 | 31.8±0.4 | 25.2±0.8 | 28.6±0.9 | 29.2±0.5 |

## 4 EXPERIMENTS

We conduct comprehensive experiments and evaluate different LLMs, using CofCA to answer the following questions: 1) Do LLMs show a performance gap between the Wikipedia-based factual MHQA datasets and our synthesized counterfactual MHQA data? 2) When inputting counterfactual questions, how do LLMs perform in terms of their reasoning ability? 3) How do sub-questions affect the performance of LLMs? 4) How do LLMs perform on reasoning chain evaluation? These investigations aim to shed light on the capabilities and limitations of LLMs when dealing with counterfactual MHQA and multi-step reasoning tasks.

### 4.1 EXPERIMENTAL SETTINGS

We evaluate LLMs on the CofCA benchmark, including 900 randomly selected data from Wikipedia-based MHQA datasets (300 QA pairs from HotpotQA (Yang et al., 2018), 300 QA pairs from 2WikiMultihopQA(Ho et al., 2020), 300 QA pairs from MuSiQue (Trivedi et al., 2021)), and 900

annotated counterfactual MHQA data (divided into 2-hop, 3-hop, and 4-hop subsets). We employ proprietary LLMs and open-source LLMs in experiments. To enhance reproducibility, we set the temperature to 0 for proprietary models, and all the experiment results are the average scores of three experiment results. We adopt the proprietary LLMs: GPT-4 (Achiam et al., 2023), GPT-3.5 (Ouyang et al., 2022), text-davinci-003, Bing Chat, GEMINI-pro (Team et al., 2023), and Open Source LLMs such as Llama 2-7b, Mistral-7b and Qwen-7b as the baselines. To decouple LLMs' internal memory and reasoning ability, and let LLMs retrieve answers from the given passage as much as possible, we design a prompt that requires LLMs to only retrieve answers based on the given context. The prompt of QA is shown in the Appendix B.

## 4.2 RESULTS

**Reasoning VS Memorization** The results of the comparison between the selected Wikipedia-based MHQA data and our CofCA can be found in Table 3 and Table 4. LLMs show a performance gap between the selected data and ours. Taking GPT-4 as an example, GPT-4 achieves high EM and $F_1$ scores (69.9 and 82.3, respectively), which are even close to well-finetuned small QA models. For our 2-hop dataset, EM and $F_1$ scores are sharply declined (53.1 in table 3 and 62.8 in table 4). For 3-hop and 4-hop datasets, GPT-4 even performs worse. Since our synthesized data is new, unprecedented knowledge, our results objectively reflect the real reasoning performance of LLMs. There is also a

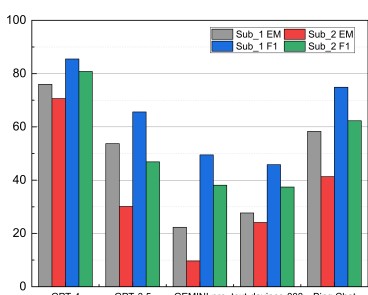

Figure 3: The performance change of $F_1$ score and EM scores when answering 2 sub-questions on the 2-hop dataset.

concern that EM may not be a dependable method due to its constraints in accurately representing real performance, particularly in scenarios where answers involve aliases or abbreviations. For example, the gold answer is "Lionel Messi" and the generated answer is "Messi". The two results should be treated the same but the EM score is not applicable in this scenario. As a result, we introduce GPT-4-turbo as the answer evaluator to measure the partial match (PM) between LLMs' generated answers and gold answers The prompts are shown in appendix B. The PM scores in table 3 and table 4 also reveal that LLMs are better at answering factual MHQA questions than counterfactual questions. In light of the results, we can find that LLMs achieve an inflated high performance on the Wikipedia-based MHQA dataset possibly because of the data contamination that leads to utilizing LLMs' memory ability rather than reasoning ability.

**Impact of Annotation on LLMs** Our annotation framework involves two stages: keywords replacement followed by paraphrasing. To demonstrate the impact of annotated passages in different stages, three experiments were conducted using the original passages, passages generated from the first stage, and passages generated from the second stage. We additionally select and annotate another 300 passages from the HotpotQA dataset. Additionally, to ensure consistency in question complexity, we only annotated 2-hop questions on both sets of passages, aligning with the HotpotQA dataset.

Table 5: Impact on LLMs' performance at different stages in the annotation framework.

| Model | HotpotQA | | 1st Stage | | 2nd Stage | |
|---|---|---|---|---|---|---|
| | EM | F1 | EM | F1 | EM | F1 |
| O1-preview | 78.35 | 86.91 | 71.55 | 78.36 | 68.18 | 74.69 |
| GPT-4 | 71.35 | 78.62 | 65.62 | 70.47 | 63.49 | 67.32 |
| GPT-3.5 | 64.58 | 72.36 | 60.74 | 66.82 | 57.94 | 60.16 |
| text | 56.67 | 61.45 | 52.43 | 55.87 | 49.65 | 52.88 |
| GEMINI-Pro | 63.28 | 68.36 | 57.94 | 60.33 | 55.46 | 58.39 |
| Bing-Chat | 61.72 | 65.16 | 53.78 | 57.69 | 51.97 | 54.37 |

Results are illustrated in Table 5, we find that: 1) there is also an obvious performance gap of LLMs between the original HotpotQA data and our two-stage data. Taking GPT-4 as an example, GPT-4 performs better on HotpotQA than our two-stage data, EM score of 71.35 and F1 score of 78.62 respectively. In the 1st stage data, GPT-4 shows a significant decrease, 65.62 EM and 70.47 F1 respectively. 2) The two stages of bias bring the different extents of performance drop. Comparing the performance of GPT-4 in stages 1 and 2, there is also an obvious drop, from 65.62 to 63.49 and from 70.47 to 67.32. It is mainly because of the data produced from the first stage although some

keywords such as named entities and noun phrases, other words, sentence structure, and internal logic are the same as the original passages. The second stage passages are totally different from the original passages. Table 10 also illustrates an example from HotpotQA that is annotated in different stages. We find that the passage in stage 1 still maintains the same sentence structure. While the passage in stage 2 is totally different from the original passage.

**Sub-QA Evaluation**    Figure 3 illustrates the performance degradation because LLMs also suffer from error propagation with the reasoning depth increasing. Figure 4 shows the performance of LLMs on the different hops of questions. According to the observation of the three figures, we find that with the hop increases, the complexity of multi-hop questions also increases, leading to the LLMs' performance decrease. When incorrectly answering the previous sub-question, the latter one will also be influenced. Consequently, the performance of Sub_Question 2 is worse than that of Sub_Question 1. Tables 13 and 14 also illustrate the sub-qa performance of LLMs on the 3-hop and 4-hop datasets in appendix D.

Table 6: Ablation study of the MHQA task, where we remove the sub-question information from the prompt and only ask LLMs to give the final answer. Here all the prompts are in a Chain-of-Thought setting.

| Setting | 2 hop | | 3 hop | | 4 hop | |
|---|---|---|---|---|---|---|
| | EM | $F_1$ | EM | $F_1$ | EM | $F_1$ |
| GPT-4 w/o Sub-Q | $43.8_{\pm0.2}$ | $65.2_{\pm0.3}$ | $41.4_{\pm0.2}$ | $61.6_{\pm0.4}$ | $38.1_{\pm0.5}$ | $48.9_{\pm0.3}$ |
| w Sub-Q | $\mathbf{53.2_{\pm0.5}}$ | $\mathbf{67.7_{\pm0.6}}$ | $\mathbf{44.5_{\pm0.2}}$ | $\mathbf{64.5_{\pm0.3}}$ | $\mathbf{42.1_{\pm0.1}}$ | $\mathbf{53.1_{\pm0.4}}$ |
| GPT-3.5 w/o Sub-Q | $34.3_{\pm0.1}$ | $51.3_{\pm0.1}$ | $32.7_{\pm0.3}$ | $48.6_{\pm0.3}$ | $31.2_{\pm0.5}$ | $41.7_{\pm0.4}$ |
| w sub-Q | $\mathbf{40.4_{\pm0.5}}$ | $\mathbf{56.9_{\pm0.4}}$ | $\mathbf{37.5_{\pm0.1}}$ | $\mathbf{50.2_{\pm0.2}}$ | $\mathbf{33.5_{\pm0.2}}$ | $\mathbf{45.9_{\pm0.6}}$ |
| GEMINI-pro w/o Sub-q | $25.2_{\pm0.5}$ | $55.2_{\pm0.7}$ | $20.8_{\pm0.6}$ | $38.7_{\pm0.4}$ | $14.1_{\pm0.2}$ | $31.0_{\pm0.4}$ |
| w sub-Q | $\mathbf{34.6_{\pm0.5}}$ | $\mathbf{64.2_{\pm0.5}}$ | $\mathbf{27.3_{\pm0.2}}$ | $\mathbf{42.1_{\pm0.2}}$ | $\mathbf{25.9_{\pm0.2}}$ | $\mathbf{33.8_{\pm0.3}}$ |
| text-davinci-003 w/o Sub-q | $24.1_{\pm0.7}$ | $48.8_{\pm0.3}$ | $22.1_{\pm0.4}$ | $45.6_{\pm0.2}$ | $20.0_{\pm0.1}$ | $42.1_{\pm0.2}$ |
| w sub-Q | $\mathbf{32.3_{\pm0.4}}$ | $\mathbf{52.7_{\pm0.4}}$ | $\mathbf{27.3_{\pm0.3}}$ | $\mathbf{46.4_{\pm0.3}}$ | $\mathbf{24.2_{\pm0.5}}$ | $\mathbf{42.8_{\pm0.7}}$ |
| Bing Chat w/o Sub-q | $37.2_{\pm0.2}$ | $52.4_{\pm0.5}$ | $33.3_{\pm0.5}$ | $44.2_{\pm0.5}$ | $29.3_{\pm0.4}$ | $38.7_{\pm0.3}$ |
| w sub-Q | $\mathbf{41.8_{\pm0.5}}$ | $\mathbf{56.8_{\pm0.5}}$ | $\mathbf{40.1_{\pm0.2}}$ | $\mathbf{48.4_{\pm0.5}}$ | $\mathbf{32.4_{\pm0.3}}$ | $\mathbf{41.3_{\pm0.3}}$ |
| O1-preview w/o Sub-q | $52.3_{\pm0.3}$ | $65.4_{\pm0.4}$ | $49.3_{\pm0.5}$ | $60.2_{\pm0.5}$ | $42.5_{\pm0.3}$ | $55.9_{\pm0.2}$ |
| w sub-Q | $\mathbf{56.7_{\pm0.3}}$ | $\mathbf{72.3_{\pm0.5}}$ | $\mathbf{40.1_{\pm0.2}}$ | $\mathbf{65.4_{\pm0.5}}$ | $\mathbf{46.4_{\pm0.3}}$ | $\mathbf{60.7_{\pm0.3}}$ |

**Joint Performance**    The joint $F_1$ $RC$ and joint EM $RC$ scores in Table 7 are the whole reasoning chain evaluation results. We find that with the increases in the reasoning chain, the performances of LLMs dropped swiftly. For example, Bing Chat gets comparable performance with GPT-4 (0.7 joint $F_1$) on answering 2 hop questions and gets a 0.9 joint $F_1$ score. However, in the 3-hop question, the joint $F_1$ $RC$ and joint EM $RC$ scores of Bing Chat are 4.2 and 8.4. In the 4-hop dataset, Bing Chat gives 4.7 joint $F_1$ $RC$, and 8.9 joint EM $RC$ scores, respectively. Since the joint performance is a negative log, the larger scores mean the worse performance on the reasoning chain. We can conclude that LLMs' reasoning ability decreases with the increases in reasoning chain length.

**Reasoning Chain Evaluation**    We calculate the proportion of the reasoning chain on the 2-hop dataset and follow the settings of (Tang et al., 2021) in calculating the percentage of correct or incorrect answers and record the results.

Table 8 shows the reasoning chain evaluation results. The green row shows the percentage of examples whose multi-hop questions can be correctly answered from the right reasoning chain. The red rows show the percentage of examples whose multi-hop questions can be correctly answered but through an incorrect reasoning chain. Among these examples, we notice that there is a low percentage of the correct final answer based on the right reasoning chain. There is also a large proportion of incorrect final answers as shown in rows 2,4,6 and 8.

Taking the results of GPT-3.5 as an example, the right reasoning chain only accounts for 13.3% although it shows a relatively high QA performance in previous tables. The percentage of incorrect reasoning chain of GPT-3.5 is 17.7% (sum of the three red rows). However, total failure cases account for 69% (sum of rows 2, 4, 6, and 8) which is substantial for the whole dataset. We conclude that LLMs only get a small proportion of the right reasoning chain and their high performance is relatively inflated due to the considerable proportion of incorrect reasoning chain.

Table 8: Categorical EM statistics (%) of sub-question evaluation for the five LLMs on our 2-hop dataset.*c* stands for *correct* and *w* stands for *wrong*. For example, the third row shows the percentage of questions where models correctly answer both 2-hop questions and the first sub-question but incorrectly answer the second sub-question. We abbreviate text-davinci-003 as text.

| $q_{sub1}$ | $q_{sub2}$ | $q$ | O1-preview | GPT-4 | GPT-3.5 | GEMINI-pro | text | Bing Chat | GPT-4o |
|---|---|---|---|---|---|---|---|---|---|
| c | c | c | 45.2 | 36.3 | 13.3 | 15.0 | 17.3 | 28.3 | 44.5 |
| c | c | w | 9.8 | 12.3 | 9.3 | 9.0 | 10.7 | 7.7 | 10.1 |
| c | w | c | 3.2 | 2.0 | 6.7 | 5.3 | 7.7 | 6.0 | 2.7 |
| c | w | w | 15.4 | 25.3 | 24.3 | 14.7 | 25.0 | 16.3 | 12.7 |
| w | c | c | 4.9 | 5.7 | 3.7 | 5.3 | 6.7 | 2.3 | 5.8 |
| w | c | w | 4.1 | 3.7 | 3.7 | 5.3 | 3.7 | 3.0 | 4.7 |
| w | w | c | 0.8 | 0.3 | 7.3 | 13.3 | 8.7 | 5. 0 | 1.4 |
| w | w | w | 16.6 | 14.3 | 31.7 | 32.3 | 30.3 | 31.3 | 18.1 |

## 4.3 IMPACT OF SUB-QUESTION

To evaluate the impact of sub-questions for LLMs, we conduct an ablation study testing the performance of answering the final answer and removing the sub-questions from prompts. All LLMs are combined with Chain-of-Thought (Wei et al., 2022) settings, in which LLMs incrementally derive a series of intermediate steps. The results, shown in Table 6, indicate that when directly asking LLM a multi-hop question and corresponding passage, the performance is much lower than that of adding sub-questions to require LLMs reasoning step-by-step. For example, computed from table 6 the performance of GPT-4 on the 2-hop dataset decreased the $F_1$ score and EM by 2.5 and 9.4 respectively. The results show the sub-questions could help LLMs improve the performance of final-QA.

Table 7: LLMs' joint performance on the whole reasoning chain. The scores are the average scores of three experiment results. The larger score means a worse performance on the whole reasoning chain.

| | 2 hop | | 3 hop | | 4 hop | |
|---|---|---|---|---|---|---|
| | $F_1$ $RC$ | EM $RC$ | $F_1$ $RC$ | EM $RC$ | $F_1$ $RC$ | EM $RC$ |
| O1-preview | 0.6 | 1.3 | 0.9 | 1.3 | 1.9 | 3.8 |
| GPT-4 | 0.7 | 1.5 | 1.1 | 1.6 | 2.3 | 4.2 |
| GPT 3.5 | 1.7 | 2.4 | 2.7 | 3.2 | 3.6 | 5.8 |
| GEMINI-pro | 2.1 | 3.9 | 4.6 | 8.7 | 5.4 | 9.5 |
| text-davinci-003 | 2.4 | 2.9 | 3.9 | 5.2 | 5.5 | 7.4 |
| Bing Chat | 0.9 | 1.9 | 4.2 | 8.4 | 4.7 | 8.9 |

## 5 ERROR ANALYSIS

We select a total of 20 incorrect final answers generated by GPT-4 from the 4-hop dataset to comprehensively illustrate how LLMs make decision on multi-step reasoning tasks. We first verify the proportion of each incorrect sub-answer and final answer. Among the 20 incorrect final answers, 9 of them are wrongly answered in the first sub-questions which leads to the incorrect final answers. While for the remaining 11, 5 of them are incorrect second sub-questions that lead to the wrong final answers. The rest of 4 are influenced by the wrong third answers and fourth answers. From this analysis, we estimate that roughly half of the incorrect final answers are incorrectly reasoning from scratch. We select 20 correct final answers generated by GPT-4 and find that about 4 of them are reasoned from incorrect reasoning chains (wrong sub-answers), revealing that LLMs also sometimes bypass the incorrect reasoning chain and get correct final answers. The insights of the whole experiment results are illustrated in Appendix E.

## 6 CONCLUSION

We presented a novel evaluation framework CoFCA for evaluating LLMs' evidence integration and multi-step reasoning capabilities by combining sub-question evaluation and counterfactual data annotation. To disentangle LLMs' memory and reasoning ability, we design a human-in-the-loop framework to synthesize counterfactual data. Our results show that, although LLMs performed relatively well on QA tasks, the performance dropped on multi-hop questions that were based on new, counterfactual knowledge. In addition, their high performances are inflated and benefit from a high proportion of incorrect reasoning chains. Our work can facilitate future research on developing faithful knowledge editing methods.

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

## A  REPRODUCIBILITY STATEMENT

To make the results and models reproducible and verifiable, we provide our full data annotation guideline, data link, implementation details, and prompts: We detail the process of data annotation in section 3.1 and the implementations are in Appendix F. All the prompts required to reproduce the results are illustrated in Appendix B.

Table 9: The prompt template of passage rewriting and question generation. We here take 2-hop data annotation as the example. [WORDS] denotes the information we should give.

---

***Prompts of NER, Noun Phrase and Adjective replacement***

---

*Prompt* Now you are a passage annotator, you need to recognize all the named entities, noun phrases, and adjectives from the given [CONTEXT], then translate the passage into Chinese and translate to English. Please output the response in JSON format {Passage: String}
*[CONTEXT]* The given context.

---

***Prompts of Question Generation***

---

*Example* One-shot example with multi-hop QA pairs, Sub-QA pairs, and passage.
*Prompt* Now you are a multi-hop question generation machine, given an example of 2 hop question and its sub-questions, sub-answers, and final answer is [2 hop question],[Sub-Questions],[Sub-Answers] and [Final Answer], you need to generate a new 2 hop multi-hop question same with the given example and its sub-questions, sub-answers and final answer from the given [Context]. Please follow the sentence structure of give examples and output the response in JSON format {2 hop question: String, sub-questions: List, sub-answers:List, final answer:String}:
*[2 hop question]* The given example of 2 hop question.
*[Sub-Questions]* The given example of sub-questions.
*[Sub-Answers]* The given example of sub-answers.
*[Final Answer]* The given example of final answer.
*[CONTEXT]* The given passage

---

***Prompts of QA***

---

*Prompt* You are a QA test machine, you need to answer the [Question] from given the [Context], and you only need to come out with the correct answer without other words. Let's think step by step, and please output the answer to the [Question] in the format of: {Final Answer: String}.
*[QUESTION]* The given question.
*[CONTEXT]* The given passage.

---

***Prompts of Partial Match Evalutaion***

---

*Prompt* You are an Answer evaluator, you need to measure the semantic similarity between [Generated Answer] and [Gold Answer], and give the score, 1 means equal, 0 means not. Some answers may have abbreviations or alias, for example, Lionel Messi is equal to Messi, Donald Trump is equal to Trump. Please only output the score 1 or 0 without any other words.
*[Generated Answer]* The LLM generated answer.
*[Gold Answer]* The ground truth.

---

# B  PROMPTS AND EXAMPLES

When evaluating large language models, prompting is a brittle process wherein small modifications to the prompt can cause large variations in the model predictions, and therefore significant effort should be dedicated to designing a painstakingly crafted perfect prompt for the given task (Arora et al., 2022; Diao et al., 2023). In this study, We investigate the performance of zero-shot on our benchmark. To eliminate the randomness, we manually select one demonstration for each task, ensuring that all tasks are covered.

We give our designed input examples for three different tasks to help readers understand our implementation, as shown in Table 9, respectively. The original and rewritten passages are shown in the table 12.

The annotated multi-hop questions are shown in the table 11.

Table 10: Examples of a passage annotated in different stages. The words in red indicate the revision part. Passage in stage 1 still has the same sentence structure with the original passage while passage in stage 2 is totally different.

| Stage | Passages |
|---|---|
| HotpotQA | Elliott Lester is an English film and television director, best known for directing the film "Blitz". He made his directing debut in 2006 with "Love Is the Drug", and his latest film, "Aftermath", was released on April 4, 2017. |
| Stage 1 | Marvin Kellaway is a British movie and radio helmsman, foremost noted for helming the movie 'Flashstrike'. He commenced his helming genesis in 2007 with 'Affection Is the Potion', and his most recent movie, 'Repercussion', was unveiled on May 3, 2018. |
| Stage 2 | Marvin Kellaway, a prominent British director in both film and radio, is best known for directing the popular movie 'Flashstrike'. He began his directing career in 2007 with 'Affection Is the Potion', and his latest film, 'Repercussion', was released on May 3, 2018. |

Table 11: Examples of annotated different question types and question hops. We emphasize keywords for their respective categories.

| Question Type | Hop | Multi-hop Question |
|---|---|---|
| Bridge | 2 hop | *When* was the actor *who* played Helen in FBC series The Murder born? |
| | 3 hop | *Who* were the learners of the people *that* was the principal violist in the Fioba Symphony Band *and* instructed music to Michard Rokney? |
| | 4 hop | *Which is later*, the birthday of Zephyr Bolt-Anderson *or* the time *that* 2060 Kingdom of Azkaban ATP Conqueror occurred in Gleeful Peak, Atlantis? |
| Comparison | 2 hop | Where is the Blue Falls Empire located *and* what products are it responsible for importing? |
| | 3 hop | *Which is later*, the opening time of Gold *or* the opening time of the Mad Book in 2006? |
| | 4 hop | *Was* the release of the movie Ocean Secrets *before or after* Echoes of Tomorrow & Victoria Wright? |

Table 12: The original passages with the rewritten passages. The first and third rows of the table are the original passages, and the second and fourth rows show the corresponding rewritten passages.

| Title | Passages |
|---|---|
| Radio City (Indian radio station) | Radio City is India's first private FM radio station and was started on 3 July 2001. It broadcasts on 91.1 (earlier 91.0 in most cities) megahertz from Mumbai (where it was started in 2004), Bengaluru (started first in 2001), Lucknow, and New Delhi (since 2003). It plays Hindi, English, and regional songs. It was launched in Hyderabad in March 2006, in Chennai on 7 July 2006, and in Visakhapatnam in October 2007. Radio City recently forayed into New Media in May 2008 with the launch of a music portal - PlanetRadiocity. com that offers music related news, videos, songs, and other music-related features. The Radio station currently plays a mix of Hindi and Regional music. Abraham Thomas is the CEO of the company. |
| Permission (African radio station) | Permission is Africa's second public FM radio station, launched on 4 August 2002. It broadcasts on 95.1 (previously 95.0 in most cities) megahertz from Baili (where it was launched in 2006), Hanwi (first launched in 2008), Shuyu, and Sadem (since 2004-09). It plays Japanese, Chinese, and folk songs. It started in Hindu in April 2007, in Beuge on 8 August 2007, and in Adler in November 2008. The Permission recently forayed into Old Business in June 2009 with the launch of a music portal - BoatPermission.com, which offers music-related news, videos, songs, and other music-related features. The Permission currently plays a mix of Japanese and folk music. Amma is the founder of the company. |
| Lights Out Paris | Lights Out Paris is the first studio album by American hip-hop artist Sims, a member of Minneapolis indie hip-hop collective Doomtree. It was released July 28, 2005, on Doomtree Records and includes guest appearances from P.O.S, Crescent Moon, and Toki Wright, among others. The album was re-released with four remixes and five songs from Sims' False Hopes Fourön vinyl in June 2015. |
| Brilliant | Brilliant is the first studio album by Australian Shout artist Allen, a member of London indie Shout collective Die. It was published on 29 October 2006 on Die Records and features guest appearances from Lucia, Lisa, and Bill, among others. The album was relisted on vinyl in July 2016, along with seven remixes and nine tracks from Allen's Right. |

## C  JOINT COMPUTATION

We here list the details of computing the joint scores on the whole reasoning chain: For example, a N-hop question and its N sub-questions, given their precisions and recalls on the MHQA ($P^{(\text{MHQA})}, R^{(\text{MHQA})}$) and the Sub-QA ($P^{(\text{sub\_}qa^1)}, R^{(\text{sub\_}qa^1)}$), ... ($P^{(\text{sub\_}qa^N)}, R^{(\text{sub\_}qa^N)}$), respectively, we calculate joint performance as:

$$P^{(\text{joint})} = P^{(\text{MHQA})} P^{(\text{sub\_}qa^1)} ... P^{(\text{sub\_}qa^N)},$$

$$R^{(\text{joint})} = R^{(\text{MHQA})} R^{(\text{sub\_}qa^1)} ... R^{(\text{sub\_}qa^N)},$$

$$\text{Joint F}_1\ RC = -\log \frac{2P^{(\text{joint})} R^{(\text{joint})}}{P^{(\text{joint})} + R^{(\text{joint})}}.$$

Table 13: The LLM evaluation on CofCA 3 hop dataset. We here measure the sub-qa task and compare the performance between each hop. $Q_i$ means the $ith$ sub-questions.

| | 3 hop | | | | | |
|---|---|---|---|---|---|---|
| | Q1 EM | Q1 $F_1$ | Q2 EM | Q2 $F_1$ | Q3 EM | Q3 $F_1$ |
| GPT-4 | $70.9_{\pm0.3}$ | $80.8_{\pm0.6}$ | $59.7_{\pm0.3}$ | $74.9_{\pm0.4}$ | $58.1_{\pm0.2}$ | $68.8_{\pm0.5}$ |
| GPT-3.5 | $43.0_{\pm0.7}$ | $56.4_{\pm0.7}$ | $38.6_{\pm0.1}$ | $49.3_{\pm0.2}$ | $29.0_{\pm0.3}$ | $40.6_{\pm0.2}$ |
| GEMINI-pro | $5.8_{\pm0.4}$ | $33.8_{\pm0.5}$ | $4.4_{\pm0.6}$ | $30.8_{\pm0.5}$ | $4.1_{\pm0.7}$ | $31.5_{\pm0.9}$ |
| text-davinci-003 | $23.3_{\pm0.8}$ | $42.4_{\pm0.3}$ | $20.5_{\pm0.4}$ | $33.7_{\pm0.3}$ | $19.5_{\pm0.5}$ | $29.6_{\pm0.6}$ |
| Bing Chat | $7.2_{\pm0.9}$ | $34.0_{\pm0.6}$ | $5.8_{\pm0.7}$ | $31.5_{\pm0.5}$ | $3.1_{\pm0.6}$ | $32.3_{\pm0.4}$ |

Table 14: The LLM performance on CofCA 4 hop dataset. We here measure the sub-qa task and compare the performance between each hop.

| | 4 hop | | | | | | | |
|---|---|---|---|---|---|---|---|---|
| | Q1 EM | Q1 $F_1$ | Q2 EM | Q2 $F_1$ | Q3 EM | Q3 $F_1$ | Q4 EM | Q4 $F_1$ |
| GPT-4 | $60.9_{\pm0.4}$ | $66.7_{\pm0.3}$ | $56.4_{\pm0.5}$ | $62.6_{\pm0.4}$ | $28.4_{\pm0.2}$ | $58.7_{\pm0.4}$ | $23.1_{\pm0.2}$ | $56.3_{\pm0.3}$ |
| GPT-3.5 | $40.7_{\pm0.4}$ | $46.9_{\pm0.3}$ | $30.1_{\pm0.2}$ | $36.3_{\pm0.2}$ | $20.2_{\pm0.1}$ | $47.2_{\pm0.5}$ | $14.7_{\pm0.4}$ | $44.8_{\pm0.2}$ |
| GEMINI-pro | $14.9_{\pm0.5}$ | $39.2_{\pm0.1}$ | $10.4_{\pm0.5}$ | $38.3_{\pm0.4}$ | $9.1_{\pm0.6}$ | $34.9_{\pm0.4}$ | $7.2_{\pm0.3}$ | $29.5_{\pm0.6}$ |
| text-davinci-003 | $19.8_{\pm0.2}$ | $39.2_{\pm0.4}$ | $19.2_{\pm0.5}$ | $30.7_{\pm0.6}$ | $18.8_{\pm0.7}$ | $28.6_{\pm0.6}$ | $18.5_{\pm0.7}$ | $27.8_{\pm0.2}$ |
| Bing Chat | $20.8_{\pm0.2}$ | $39.4_{\pm0.4}$ | $16.9_{\pm0.2}$ | $37.1_{\pm0.3}$ | $6.2_{\pm0.5}$ | $35.8_{\pm0.4}$ | $5.5_{\pm0.7}$ | $35.1_{\pm0.3}$ |

where the Joint $F_1$ $RC$ means the joint $F_1$ performance of the reasoning chain.

Given their EM scores on the MHQA ($EM^{(\text{MHQA})}$) and the Sub-QA $EM^{(\text{sub\_}qa^1)}$), ... $EM^{(\text{sub\_}qa^N)}$.

$$\text{Joint EM } RC = -\log \frac{2EM^{(\text{MHQA})}, ...EM^{(\text{sub\_}qa^N)}}{EM^{(\text{MHQA})}+, ...EM^{(\text{sub\_}qa^N)}}.$$

where the Joint EM $RC$ means the joint EM performance of the reasoning chain.

## D  PERFORMANCE ANALYSIS

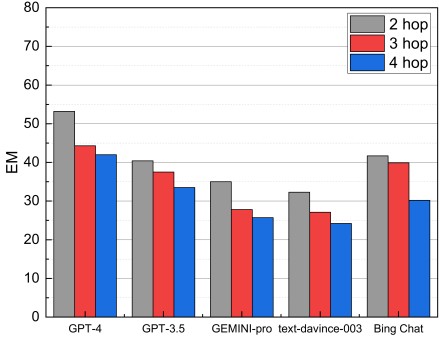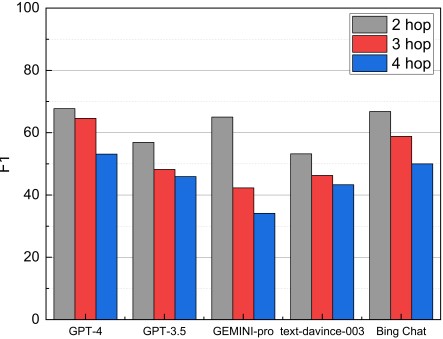

Figure 4: The performance change of EM and F1 scores when answering from 2 hop questions to 4 hop questions.

As the quantitative complementary of Sub-QA evaluation, we here list the results of LLMs' performance on 3-hop and 4-hop datasets. The LLMs' reasoning performance dropped dramatically, e.g. in table 13, GPT-4 achieves 70.9 EM and 80.8 $F_1$ scores on sub-question1 but only gets 59.7 EM, 74.9 $F_1$, and 58.1 EM, 68.8 $F_1$ scores on sub-question2 and sub-question3 respectively. In table 14, we further find that when answering 4 hop questions, the results show a cliff-like descent from sub-question2 to sub-question3, especially GPT-3.5 gets 46.9 $F_1$ in sub-question2 but drop to 36.3 $F_1$ score in sub-question3.

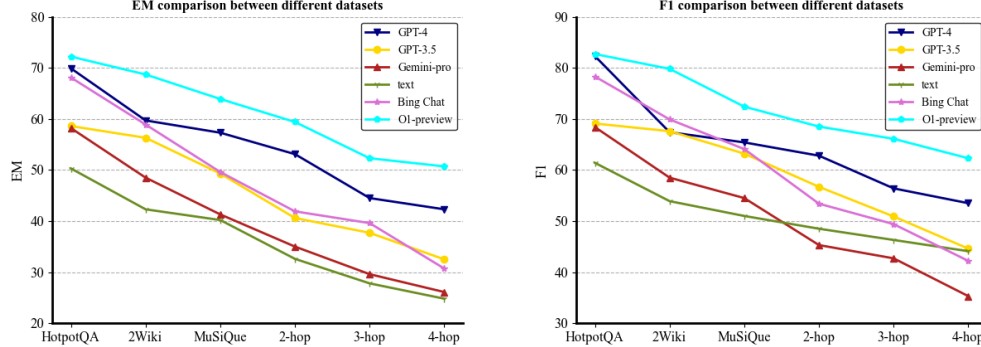

Figure 5: Performance gap between Wikipedia-based factual multi-hop QA datasets and our 2-hop, 3-hop, and 4-hop counterfactual MHQA data of table 3 and table 4. The line charts reveal that LLMs show an obvious performance gap between previous datasets and CofCA.

Table 15: Error comparison on factual and counterfactual data on different question types.

| Question Type | Data Type | Wrong | Correct |
|---|---|---|---|
| Comparison | factual | 6 | 14 |
|  | counterfactual | 11 | 9 |
| Bridge | factual | 4 | 16 |
|  | counterfactual | 8 | 12 |

We randomly sampled 40 factual data (20 bridge questions and 20 comparison questions) from the HotpotQA dataset and 40 counterfactual data (20 bridge questions and 20 comparison questions) from our 2-hop data. We count the number of wrong and correct answers given by GPT-4 on different question and data types. Table 15 illustrates the results on our selected 80 data. We find that Although GPT-4 is worse on counterfactual data than factual data, GPT-4 is better at answering bridge-type questions than comparison questions. We further analyze the error categories on the errors. We

Table 16: Statistics of different error categories

| Data types | Factual | | Counterfactual | |
|---|---|---|---|---|
| Question Types | Bridge | Comparison | Bridge | Comparison |
| Reasoning failures | 3 | 5 | 5 | 5 |
| Retrieval Errors | 1 | 1 | 1 | 3 |
| Contradictions | 0 | 0 | 2 | 3 |

categorize errors into three types: reasoning failures, evidence retrieval errors, and contradictions. We compare the number of different error categories and analyze their relative prevalence in factual versus counterfactual datasets. In factual data, since all knowledge of passages is from the real world, there are no contradictions. However, in counterfactual data, there are 5 contradiction errors which means the knowledge is contradicted to LLMs' knowledge. E.g. the question: "When did Africa's second public FM radio station launch?" In Wikipedia, the second radio station is "Rádio Nacional de Angola (RNA) " which was launched in 1951, while in our counterfactual passage, the answer is "Permission", launched in 2002.

## E  INSIGHTS OF LLM MULTI-STEP REASONING EVALUATION

Drawing from the above experimental results, we draw the conclusions:

**Exact Matching**  While exact matching is a simple and effective method for MHQA evaluation, it struggles with issues when the answers have abbreviations or other expressions. For example, in our synthesized counterfactual MHQA data, if the golden answer to the question "When did Africa's second public FM radio station launch?" is "2002" and the generated answer of GPT-4 is "4 August 2002", the exact match can not be computed accurately. All the answers generated by LLMs have this issue. Thus, It is urgent to propose a more universal QA evaluation score in LLMs' reasoning performance evaluation.

**Multi-step Reasoning**  Although the experiment results demonstrate that LLMs can perform multi-step reasoning ability to a certain extent, they remain sensitive to prompts and the impact of additional contexts, especially the sub-questions. Providing sub-questions as additional information into prompts can help guide LLMs to reason in the correct direction and show a relatively strong performance.

**Reasoning chain Evaluation (Joint F1 RC and Joint EM RC in this work)**  The advantage of our reasoning evaluation method is we jointly consider the sub-QA performance and final-QA performance when LLMs bypass the incorrect reasoning chain and achieve a correct final answer, the scores remain very low. However, this evaluation method is easily influenced by the LLMs' performance on the first sub-questions, since the answer order is sequential. if the first sub-question is incorrectly answered, the following sub-questions and the final question will also be influenced, leading to a very low score. How to answer sub-questions more correctly remains exploration.

## F  IMPLEMENTATION DETAILS

For proprietary models, we employ official APIs to interact with exclusive LLMs and prompts are well-defined. For open-source models, all experiments are conducted on 8 A100 GPUs.

## G  LIMIATATIONS

In this paper, we focus on the evaluation of LLMs' real multi-step reasoning ability on our annotated counterfactual MHQA data. Although LLMs show an obvious performance gap between previous factual MHQA datasets and our dataset, the data size of our dataset still remains improved. The Exact Match (EM) for reporting QA performance still faces challenges, because EM does not report LLMs' real performance due to the variation in the expression of the answers.

