# OpenReview forum: "CofCA: A STEP-WISE Counterfactual Multi-hop QA benchmark"
_ICLR.cc/2025/Conference — ICLR 2025 Poster_

### Official Review · Reviewer_gcH3 · 2024-10-30

**Soundness:** 3
**Presentation:** 3
**Contribution:** 3
**Rating:** 8
**Confidence:** 4

**Summary:**

The paper is designed to evaluate large language models (LLMs) on multi-hop counterfactual reasoning abilities. CofCA generates counterfactual contexts that LLMs likely haven’t encountered during pre-training, and ask questions based on the counterfactual contexts.  It also emphasizes sub-question evaluation for each reasoning step.

The evaluation in this paper reveals that models like GPT-4 show inflated scores in standard QA due to these contamination issues, while performance drops on CofCA due to its reasoning on counterfactual contexts, highlighting the gap between LLM memory and genuine reasoning.

**Strengths:**

1. CofCA minimizes data contamination by creating a cleaner benchmark for evaluating LLM reasoning.

2. The diverse counterfactual and factual datasets allow researchers to observe performance discrepancies, adding depth to LLM evaluation.

3. The constructed dataset is clearly different from existing counterfactual QA benchmarks, including distentQA and IfQA. The multi-hop challenge lacks of enough study in counterfactual setting.

**Weaknesses:**

1. More open-source LLMs should be evaluated on the benchmark, like llama-3 or 13B-level models.

2. OpenAI o1 was evaluated in the paper, it shows using long reasoning-chains could help a lot on this reasoning tasks, and more insights on comparing o1 reasoning with 4o reasoning could be insightful.

**Questions:**

As given in the weakness.

---

> ### Author Response · Authors · 2024-11-21
> **Response to Reviewer gcH3**
>
> ***Response 1:***
> We sincerely appreciate the reviewer’s constructive feedback and valuable suggestions. Below, we address the concerns raised in the review point by point:
> We here compare the performance of several open-source and commercial LLMs on the 2-hop of the CofCA dataset and the HopotQA dataset.
> | Models | HotpotQA | HotpotQA |  CofCA | CofCA |
> |--------|-----|------|----|----|
> |        | EM | F1 | EM | F1 |
> | GPT-4 turbo | 69.2 | 78.7 | 56.8 | 60.6 |
> | GPT-40 | 71.5 | 83.5 | 57.5 | 62.2 |
> | Claude | 53.8 | 67.3 | 49.6 | 64.3 |
> | LlaMa 3-8B | 40.8 | 57.1 | 34.5 | 52.8 |
> | LlaMa 2-13B | 43.7 | 60.5 | 39.6 | 55.4 |
> | Deepseek 7B Chat | 39.8 | 56.3 | 36.8 | 54.8 |
>
> ***Table 1. LLMs on CofCA and HotpotQA datasets.***
>
> The selected LLMs also exhibit varying extent of performance degradation on HotpotQA and our CofCA.
> Due to the time limit, we will add the full experiment results including 2WikiMultihopQA and MuSiQue datasets in the final version of our paper.
>
> Besides, we compare the reasoning chain performance of GPT-4o and O1-preview:
>
> |Sub Q1|Sub Q2| Q|O1-preview | GPT-4o |
> |--------|--------|----|------------|--------|
> | c   | c  | c  | 45.2 | 44.5   |
> | c   | c  | w  | 9.8  | 10.1   |
> | c   | w  | c  | 3.2  | 2.7    |
> | c   | w  | w  | 15.4 | 12.7   |
> | w  | c   | c  | 4.9  | 5.8    |
> | w  | c   | w  | 4.1  | 4.7    |
> | w  | w   | c  | 0.8  | 1.4    |
> | w  | w   | w  | 16.6 | 18.1   |
> ***Table 2. Reasoning Chain performance of LLMs.***
>
> Analysis:
> GPT-4o achieves the best performance of all models, but still a large margin between HotpotQA and the 2-hop subset. Due to the time limit, we will add the full experiment results on 3-hop and 4-hop subsets with the model performance and update Tables 3 and 4.
>
> Compared with O1-preview. GPT-4o achieves comparable performance on the correct reasoning chain (45.2 vs 44.5 in the first row). However, GPT-4o is easier to be influenced by incorrect sub-answers and performs worse on incorrect reasoning chains. (16.6 vs 18.1 in the last row, 4.9 vs 5.8 in the fifth row). When correctly answering the first sub-questions, GPT-4o is harder to be influenced when the answer to the second sub-question is wrong (15.4 vs 12.7 in the fourth row.) Accordingly, we added the performance of GPT-4o in Table 8.

---

> > ### Comment · Reviewer_gcH3 · 2024-11-25
> >
> > Thanks for your response, I am still leaning to accepting the paper, and will maintain my score.

---

### Official Review · Reviewer_cSw3 · 2024-10-30

**Soundness:** 3
**Presentation:** 3
**Contribution:** 2
**Rating:** 5
**Confidence:** 4

**Summary:**

This paper introduces a new dataset called CofCA, designed to effectively assess LLMs' capability in counterfactual multi-hop question answering. The authors generate the dataset by starting with Wikipedia passages and applying techniques such as entity replacement and back-translation to create question-answer pairs. The models are then evaluated at both the final-answer level and the sub-question level. To ensure the quality of the dataset, the authors perform thorough quality checks and include human-in-the-loop validation throughout the dataset construction process.

**Strengths:**

1. The paper is well-written and easy to follow, presenting the content in a clear and accessible manner.

2. The authors demonstrate an awareness of the limitations of using Exact Match (EM) and F1 scores when evaluating the output of large language models (LLMs). To address this, they introduce an additional Partial Match metric, which helps mitigate the issue and provides a more nuanced evaluation.

**Weaknesses:**

1. One major drawback of the paper is that, while it claims to assess LLMs' true reasoning abilities, it primarily reports the accuracy of different sub-questions without fully addressing whether the models are capable of reasoning holistically. For a more comprehensive evaluation of reasoning abilities, other factors should also be considered, such as whether the model is able to effectively decompose the problem into meaningful or correct sub-questions, rather than treating them as isolated tasks.

2. The quality of the dataset is also in question. Currently, the correctness and informativeness of the dataset are evaluated using GPT-4. However, as the paper itself shows, GPT-4 struggles to correctly answer these questions, which suggests that it may not be fully reliable for assessing the quality of the dataset. This raises concerns about the robustness of the dataset validation process.

3. There is further concern about ensuring that the generated questions are non-trivial. As shown in Table 9, the prompt generates the multi-hop questions and the sub-questions, along with their answers, simultaneously. This raises a natural concern that the generated multi-hop questions may either be independent of these sub-questions or follow a single pattern (e.g., q1->q2->q3), leading to a lack of diversity in question structure. The paper does not address or discuss this potential limitation.

4. When benchmarking models, the paper lacks evaluation on more recent and advanced models, such as GPT-4 Turbo, GPT-4O, and Claude series, as well as newer open-source models like LLaMA 3 or Deepseek. Including these models would provide a more up-to-date and comprehensive analysis of the current state-of-the-art in LLMs.

**Questions:**

1. Could you provide a more intuitive explanation for the **JOINT COMPUTATION** metric mentioned in Appendix B? An example or simplified breakdown might help readers understand its significance and how it is calculated in practical terms. This would not only make technical details more accessible but also allow readers to better appreciate its role in evaluating model performance.

---

> ### Author Response · Authors · 2024-11-21
> **Response to Reviewer cSw3**
>
> ***Response weakness 1:***
>
> We thank the reviewer for your constructive feedback, thoughtful evaluation, and recognition of the strengths of our work. We are pleased that the clarity of our paper, the introduction of the Partial Match metric, and our efforts to address limitations in traditional evaluation metrics were appreciated. Below, we address the concerns raised in the review point by point:
> We agree with the reviewer’s observation that our paper primarily focuses on sub-question accuracy and that reasoning abilities involve more than treating sub-questions as isolated tasks. To address this concern:
>
> We conducted a new experiment based on the suggestions, where we randomly selected 100 data 2-hop dataset and 100 data from the HotpotQA dataset. We require LLMs to generate sub-questions based on the given multi-hop question and passage and compute the BLEU and Rouge-L scores.
>
> | Models       | HotpotQA       |      HotpotQA  | 2-hop          |       2-hop    |
> |--------------|----------------|----------------|----------------|----------------|
> |              | BLEU           | Rouge-L        | BLEU           | Rouge-L        |
> | GPT-4        | 27.5           | 58.4           | 26.6           | 56.9           |
> | GPT-3.5      | 22.1           | 50.6           | 20.5           | 47.1           |
> | O1-preview   | 33.2           | 64.7           | 30.4           | 52.2           |
> | GEMINI-pro   | 23.8           | 47.2           | 22.1           | 43.2           |
> | text         | 21.5           | 46.3           | 19.8           | 45.5           |
> | Bing Chat    | 24.6           | 51.2           | 22.4           | 48.7           |
> ***Table 1. Question Decomposition performance of LLMs***
>
> Analysis:
> The question decomposition ability of LLMs has almost not decreased, which is mainly because LLMs just need to understand the question structure and all evidence given in passages, thus emphasizing the holistic nature of reasoning.
>
> ***Response weakness 2:***
> The reviewer raises a valid point about using GPT-4 for dataset validation while also evaluating GPT-4 on the dataset. To address this:
> Human Verification Process:
> While GPT-4 assisted in generating the dataset, all counterfactual paragraphs and question-answer pairs underwent manual verification by trained annotators to ensure correctness, informativeness, and alignment with the counterfactual modifications. We randomly selected 300 data (100 from the 2-hop subset, 100 from the 3-hop subset, and 100 from the 4-hop subset) and asked a third expert to review the data quality.
> The third expert checks: 1) grammar issue; 2) whether Exists in the real world or not; 3)Answerable to the passages and QA pairs. We show the third expert review results as follows:
>
> | Quality Issues    | 2-hop | 3-hop | 4-hop |
> |-------------------|-------|-------|-------|
> | Grammar Issue     | 2%    | 5%    | 6%    |
> | Existence         | 0     | 0     | 0     |
> | Answerability     | 1%    | 1%    | 3%    |
> ***Table 2. Data Quality***
>
> Diversity Validation:
> We include an additional analysis comparing the diversity of question structures and reasoning chains, demonstrating that the dataset contains non-trivial multi-hop reasoning challenges beyond simple patterns.
>
> | Type       | Question                                                                 | Structure          |
> |------------|---------------------------------------------------------------------------|---------------------|
> | Bridge     | What's the 45th Graple Group and when was it inactivated?                | compound sentence   |
> | Bridge     | What's the name of the band built by the man who is an Argentina Gaffer? | relative clause     |
> | Bridge     | Fly forests are a vegetation type endemic to which country?              | interrogative sentence |
> | Comparison | Horrible Husband and Bus, Who was founded earlier?                       | interrogative sentence |
> | Comparison | Which one is released first, Substance or Thunder Games?                | compound sentence   |
> | Comparison | What are the nationalities and professions of Ally Lahr and Dilar Jabilina? | nominal clause      |
> ***Table 3. Question examples and sentence structure.***
>
> Although questions are the same question type, their sentence structures are different.

---

> ### Author Response · Authors · 2024-11-21
> **Response to Reviewer cSw3**
>
> ***Response weakness 3***：
>
> The reviewer highlights a potential concern about whether the generated multi-hop questions and sub-questions exhibit sufficient diversity. We recognize the importance of this point and offer the following clarifications and plans:
>
> The multi-hop question and corresponding sub-questions are reviewed by humans which means it must be related to the annotated passages (Answerability), which ensures the sub-questions are dependent on multi-hop questions.
> our questions mainly contain two question types, bridge and comparison. The bridge questions are single pattern q1-q2-q3. While the comparison question is not a single pattern, the answer final question relies on the comparison of all sub-answers.
>
> Here we list two examples of our data:
> Bridge Question: "When did Africa's second public FM radio station launch?"
> Sub-Question 1:"What is Africa's second public FM radio station?"
> Sub-Question 2: "When did Permission launch?"
>
> Comparison Question:"Which is later, the opening time of Gold or the opening time of the Mad Book?"
> Sub-question 1: "When is the opening time of Gold?"
> Sub-question 2: "When is the opening time of Mad Book?"
> Sub-question 2 :"Which year is later between 1935 and 2006?"
>
>
> ***Response weakness 4:***
> We appreciate the suggestion to benchmark CofCA on more recent and advanced models. We plan to address this by extending benchmarking:
> | Models           | HotpotQA       | HotpotQA       | CofCA          |   CofCA        |
> |------------------|----------------|----------------|----------------|----------------|
> |                  | EM             | F1             | EM             | F1             |
> | GPT-4 turbo      | 69.2           | 78.7           | 56.8           | 60.6           |
> | GPT-4o           | 71.5           | 83.5           | 57.5           | 62.2           |
> | Claude           | 53.8           | 67.3           | 49.6           | 64.3           |
> | LLaMA 3-8B       | 40.8           | 57.1           | 34.5           | 52.8           |
> | LLaMA 2-13B      | 43.7           | 60.5           | 39.6           | 55.4           |
> | Deepseek 7B Chat | 39.8           | 56.3           | 36.8           | 54.8           |
>
> Analysis:
> GPT-4o achieves the best performance of all models, but still a large margin between HotpotQA and 2-hop. Due to the time limit, we will experiment with the model performance on the whole dataset and update Tables 3 and 4.
>
> ***Response Question: Details of computing Joint Score***
>
> The reviewer requested a more intuitive explanation of the JOINT COMPUTATION metric. Here’s an outline of how we clarify this in the revised submission:
>
> We provide an example illustrating how the metric works in practice:
> The multi-hop question: "When did Africa's second public FM radio station launch?"
> Sub-Question 1: "What is Africa's second public FM radio station?"
> Sub-Question 2: "When did Permission launch?"
> Sub-answer 1: "Permission"
> Sub-answer 2: "4 August 2002"
> Final Answer: "4 August 2002"
>
> We first compute the F1 score and EM score of each sub-question and final question. Then we compute the joint score based on each F1 score and EM score.
>
> Since in the reasoning chain, once the first sub-question is answered incorrectly, the subsequent questions are prone to be wrongly answered. The joint score is a negative log of F1 scores and EM scores. Consequently, The higher the score, the lower the performance of the reasoning chain.
>
> By emphasizing the dependency between sub-question answers and the final answer, this metric helps expose reasoning shortcuts and incomplete or flawed reasoning processes.

---

> > ### Comment · Reviewer_cSw3 · 2024-11-25
> >
> > Thanks for the detailed response. I decide to keep my rating unchanged.

---

### Official Review · Reviewer_royr · 2024-11-02

**Soundness:** 3
**Presentation:** 2
**Contribution:** 2
**Rating:** 5
**Confidence:** 4

**Summary:**

The paper introduces CofCA, an evaluation benchmark for multi-hop reasoning abilities of LLMs over counterfactual data. CofCA is created by (1) LLMs alter Wikipedia passages by replacing keywords with counterfactual information, and (2) LLMs generate multi-hop questions over the modified passages. Both steps are manually verified by human annotators, and the benchmark includes 900 multi-hop questions from 300 passages. Experimental results show a gap between performance on the counterfactual CofCA and existing multi-hop QA datasets. Additionally, the paper shows that the low performance is due to reasoning failures.

**Strengths:**

- A counterfactual multi-hop benchmark can be helpful to understand the ability of LLMs to perform multi-hop reasoning. The benchmark will be publicly available in addition to the annotation guidelines and implementation details.

- The paper features extensive quantitative analysis examining the effect of the annotation pipeline and the performance of sub-questions amongst others.

**Weaknesses:**

- Given the examples presented in the paper, I was not fully convinced about several aspects of the collected data. Mainly, how do you ensure that the counterfactual information does not contradict real-world facts? Also, both examples in the main paper seem to have shortcuts, e.g., there is only one position in the example in Fig.1, and only one designer in the example in Fig. 2.

- Qualitative analysis - the analysis presented in Section 5 did not fully address the concerns above, and is limited to 20 correct/incorrect questions with GPT-4. I think the paper could benefit from a thorough analysis regarding the quality of the generated data examining contradiction and shortcuts and verifying that the main claims of the paper (multi-hop QA is more challenging on counterfactual information) is supported. For example, it could have been interesting to have an error analysis that compares different error categories between factual and counterfactual questions.

- Presentation - the presentation in several parts of the paper was a bit hard to follow. For example, in Fig.1, the text is very small and there is an error (the bottom-right box the question is about the coach and the answer is Point Guard). In Tab.2, the ‘Whole data property’ is not the sum of the other columns (I assume this is due to the factual questions sampled from other datasets?). The main results in Section 4.2 require comparing between information presented in different tables (Tables 3 and 4). Table 8 is referenced before Table 7 but presented afterwards.

**Questions:**

See weaknesses, mainly:
- Does the counterfactual information contradicts real-world facts? If so, isn't some degradation in performance expected?
- How often are shortcuts in the collected examples?
- What qualitative analysis did you consider that might strengthen the main claims of the paper?

---

> ### Author Response · Authors · 2024-11-21
> **Response to Reviewer**
>
> ***Response weakness 1:***
>
> We sincerely appreciate the reviewer’s constructive feedback and valuable suggestions. Below, we address the concerns raised in the review point by point:
>
> 1) Counterfactual modifications were generated by GPT-4 and subjected to manual verification by annotators to ensure internal coherence and consistency with the provided context. Our instructions explicitly required annotators to confirm that counterfactual information introduced plausible yet alternative scenarios without conflicting with factual grounding. For example, in Figure 1 d, no such coach that serves the Chinese national basketball team and was born in 1997. Once we were given such counterfactual passages, LLMs could only generate answers based on the given passages.
>
> 2) While some degradation in performance is expected when models encounter counterfactual scenarios, our findings indicate that the extent of the performance gap between CofCA and factual datasets cannot be solely attributed to contradictions. Instead, as our quantitative and qualitative analyses highlight, this gap primarily stems from reasoning failures rather than mere unfamiliarity with the counterfactual context.
>
>
> ***Response weakness 2:***
>
> The reviewer suggests that the qualitative analysis in Section 5 could be more thorough. We appreciate this feedback and provide the following modifications:
>
> 1) Expanded Error Analysis:
> We will enhance our error analysis to explicitly compare error categories between factual and counterfactual questions. This will provide deeper insights into the specific challenges posed by counterfactual reasoning.
> For example, we categorized errors into types such as reasoning failures, evidence retrieval errors, or contradictions and analyze their relative prevalence in factual versus counterfactual datasets.
> 2) Analysis of Contradictions and Shortcuts:
> As part of the expanded qualitative analysis, we examine whether any contradictions or shortcuts in the dataset affect model performance. This will directly address the reviewer’s concerns and reinforce the claims made in the paper.
>
> In particular, we randomly sampled 40 factual data (20 bridge questions and 20 comparison questions) from the HotpotQA dataset and 40 counterfactual data (20 bridge questions and 20 comparison questions) from our 2-hop data.  The answer is generated by GPT-4.
> | Question Type | Data Type    | Wrong | Correct |
> |---------------|--------------|-------|---------|
> | **Comparison**| factual      | 6     | 14      |
> |               | counterfactual| 11    | 9       |
> | **Bridge**    | factual      | 4     | 16      |
> |               | counterfactual| 8     | 12      |
>
> ***Table 1 Error Analysis on Question Types.***
>
> The total number of errors in factual data is 10, total errors in counterfactual data is 19.
> Analysis: Although GPT-4 is worse on counterfactual data than factual data, GPT-4 is better at answering bridge-type questions than comparison questions.
>
> We also analyze the error types of the errors:
> | Data Types          | Factual   | Factual     |Counterfactual    | Counterfactual    |
> |---------------------|--------|----------|--------|-----------|
> | Question Types      | Bridge | Comparison | Bridge | Comparison |
> | Reasoning Failures  | 3      | 5          | 5      | 5          |
> | Retrieval Errors    | 1      | 1          | 1      | 3          |
> | Contradictions      | 0      | 0          | 2      | 3          |
> ***Table 2. Error Types.***
>
> In factual data, since all knowledge of passages is from the real world.
> There are no contradictions. However, in counterfactual data, there are 5 contradiction errors which means the knowledge is contradicted to LLMs’ knowledge.
>
> Accordingly, we added the experiment results and analysis in Appendix D, words in red.
>
> ***Response weakness 3:***
>
> We made the following improvements in the revised manuscript:
> Figure and Table Revisions:
> 1) Figure 1: Increase the font size, correct the errors regarding the question-answer mismatch, and provide a clearer explanation in the caption.
> 2) Tables 3, 4, 7, and 8: Reorganize the presentation to ensure sequential referencing and easier cross-comparison of results. For example, we will consolidate key insights into a unified table or accompanying visualizations for better clarity.
> Clarifying Table 2:
> 3) We explicitly explain why the "Whole data property" column is not the sum of the others (e.g., due to factual questions sampled from other datasets). This will remove ambiguity and improve interpretability. We added the explanation of total data in section 3.1, paragraph “Dataset Analysis and Statistics”, words in red.
> 4) We reorganize Sections 4.2 and 5 to make the experimental results and analyses easier to follow. This includes adding transition sentences, summarizing key findings more explicitly, and aligning section content with the overarching narrative.

---

> > ### Comment · Reviewer_royr · 2024-11-25
> >
> > Thank you for your response! I have some follow-up questions, and perhaps I am missing something.
> >
> > - Which analysis examines whether the effect of shortcuts on model performance?
> > - Doesn't the new analysis shows that the delta in reasoning failures between the counterfactual and factual setting is relatively small (5 errors for counterfactuals in each setting vs 3-5 for factual), while the delta in contradictions is larger (2-3 for counterfactuals vs none for factual)? Additionally, this seems like relatively small numbers for drawing conclusions (other than that a ~10-15% decrease is expected for counterfactual data due to contradictions).

---

> > > ### Author Response · Authors · 2024-11-26
> > > **Response to Reviewer**
> > >
> > > Thank you for the comments.
> > > First of all, the analysis of shortcuts:
> > > In Table 8, we discussed the shortcuts, which means LLMs sometimes get incorrect intermediate answers but reach the correct final answers.
> > >
> > > The red rows of the table 8, illustrate the proportion of LLMs’ shortcuts in answering sub-questions.
> > > We analyze it in section 4.2 page 9.
> > >
> > > Secondly, since we just selected 40 data from our CofCA dataset, and 40 data from the HotpotQA dataset, the experiment results can not fully reflect the real performance drop between counterfactual and factual data.
> > > In the final version of our paper, we will select 100 data from CofCA and 100 data from HotpotQA and conduct the error analysis again.

---

> > > > ### Comment · Reviewer_royr · 2024-11-29
> > > >
> > > > Thank you for your response. I don't have any follow-up questions.

---

### Official Review · Reviewer_6xmP · 2024-11-04

**Soundness:** 3
**Presentation:** 3
**Contribution:** 3
**Rating:** 6
**Confidence:** 4

**Summary:**

This paper proposed a new dataset, CofCA for evaluating LLM’s counterfactual multi-hop reasoning ability, which focuses on both final answer evaluation and reasoning step evaluation. The dataset construction involves 2 steps, 1) the sampled Wikipedia paragraphs are sent to GPT-4 for entity replacement and paraphrase, and then back-translation is done to further modify the paragraphsm and then these paragraphs are manually checked 2) GPT-4 is then used to generate QA pairs based on the newly generate paragraphs. Here GPT-4 is asked to generate both overall multi-hop QA pair and subquestion answer pairs, and the resulting data is also manually verified.
The authors then conducted experiments on both newly generated data and HotpotQA, 2WikiHop and Musique, using both closed-sourced models and open-source models. The results show that all LLMs suffer significant performance drop on counterfactual questions and the joint performance which considers both final answer and reasoning process correctness can be even worse, exposing the weakness of current LLMs.

**Strengths:**

1. CofCA could be a valueable resource for studying LLM’s true reasoning ability. Due to its counterfactual nature, the LLMs can no longer use their internal knowledge for finding reasoning shortcut, hence the model have to reason over provided context. Also, the annotation of intermediate sub question answer pairs can also enable examination of the reasoning process.

2. The experiments and analysis are very thorough. The results show a clear trend of performance vs question complexity and the analysis on the reasoning process further exposes the weaknesses of the SOTA LLMs.

**Weaknesses:**

My main concern is the scale of the passages set when constructing the data. For previous datasets such as. HotpotQA and 2WikiHop, they usually construct questions using multiple passages to ensure that the answer have to be deducted from multiple pieces of evidence. However, it seems that for CofCA, the questions are generate using only a single passage? If this is the case, how can you make sure the generated multi-hop QA pairs are fully grounded to the passage?

**Questions:**

See weaknesses

---

> ### Author Response · Authors · 2024-11-21
> **Response to Reviewer 6xmP**
>
> We sincerely thank the reviewer for your thoughtful feedback and detailed analysis of our submission. Below, we address the concerns raised in the review point by point:
>
> ***Multi-Hop Grounding in CofCA:***
>
> Due to the space limit, the examples in Figures 1 and 2 are presented in single-passage QA, which is utilized to illustrate the difference between our evaluation framework and previous work.
> However, the CofCA questions are derived from multi-paragraphs (10 paragraphs, 38.42 sentences on average) of our dataset ensures that answering requires reasoning across multiple hops within the passage.
> We follow the passage settings of HotpotQA and 2WikiMultihopQA datasets.
> The counterfactual modifications, paraphrasing, and back-translation operations substantially alter the semantic structure of the passages. As a result, the questions often demand reasoning over distinctively transformed segments across paragraphs.
> Furthermore, the intermediate subquestion-answer pairs explicitly ensure that the multi-hop reasoning is both well-structured and fully grounded. We manually verified these pairs to validate their alignment with the generated questions and to avoid spurious correlations.
>
> ***Revisions:***
>
> We added the explanation of data annotation in section 3.1, paragraph “QA-pair Annotation and Checking”, words in red.
> The reply is based on our understanding of “multiple pieces of evidence”. If you have further questions, we would be happy to answer them.

---

> > ### Comment · Reviewer_6xmP · 2024-11-25
> >
> > Thank you for your response, I will keep my current ratings.

---

### Meta-Review · Area_Chair_YKo3 · 2024-12-20

**Metareview:**

There is a discrepancy among the review scores. Yet it is worthy to note that the two reviewers who gave 5 did not participant in the after-rebuttal discussion.

I have personally read this paper and find it to be a valuable contribution. It tackles an important challenge in our field: how to disentangle the evaluation of LLMs’ reasoning abilities from their reliance on internal memory. The dataset construction process proposed in the paper is well-conceived, and the experimental results support its validity. Despite the evaluation being limited to specific scenarios and some unresolved issues raised by the reviewers, I believe the paper makes meaningful progress toward the long-standing goal of evaluating pre-trained models effectively.

That said, I concur with the reviewers on several notable weaknesses that remain unresolved:

- The paper lacks additional insights, particularly regarding the decomposition of questions.
- There is insufficient clarity on how the counterfactual information is ensured to not contradict real-world facts.

Furthermore, the absence of human performance benchmarks in the results table is a notable omission. Including a comparison involving human participants unfamiliar with the dataset construction would provide a more comprehensive evaluation.

**Additional Comments On Reviewer Discussion:**

A summary of the unsolved points after rebuttal:

- The paper lacks additional insights, particularly regarding the decomposition of questions (Reviewer cSw3)
- There is insufficient clarity on how the counterfactual information is ensured to not contradict real-world facts (Reviewer royr)

---

### Decision · Program_Chairs · 2025-01-22

Accept (Poster)